# Differences in self-perception of productivity and mental health among the STEMM-field scientists during the COVID-19 pandemic by sex and status as a parent: A survey in six languages

**Seulkee Heo**[1]*, **Pedro Diaz Peralta**[1,2], **Lan Jin**[3], **Claudia Ribeiro Pereira Nunes**[1,4], **Michelle L. Bell**[1]

1 School of the Environment, Yale University, New Haven, Connecticut, United States of America,
2 Administrative Law Department, School of Law, Universidad Complutense de Madrid, Madrid, Spain,
3 School of Public Health, Yale University, New Haven, Connecticut, United States of America, 4 Graduate Program in Law, School of Law, Federal University of Amazon, Manaus, Amazonas, Brazil

* seulkee.heo@yale.edu

**Data Availability Statement:** The data contain potentially sensitive health information, and the

## Abstract

The COVID-19 pandemic has caused unprecedented challenges for working conditions for scientists, but little is known for how the associations of these challenges with scientists' mental health and productivity differ by sex and status as a parent. This online survey study in six languages collected data from 4,494 scientists in Science, Technology, Engineering, Mathematics, and Medicine fields across 132 countries during October–December 2021. We compared the type of challenges for work, changes in work hours, and perception in productivity during the pandemic by sex and status as a parent (children <18 years living at home). Regression analyses analyzed the impacts of changed working conditions and work-life factors on productivity and mental health. We found that the percentage of participants with increased work hours was the highest in female participants, especially without children. Disproportionately higher increases in work hours were found for teaching and administration in women than men and for research/fundraising in non-parent participants than parent participants (p-value<0.001). Female participants were more concerned about the negative impacts of the pandemic on publications and long-term career progress, and less satisfied with their career progress than their male counterparts. There were differences in the type of institutional actions for the pandemic across study regions. The identified obstacles for work and home-life factors were associated with higher risks of experiencing depression, anxiety, and stress. Decision makers should consider the gender differences in the pandemic's adverse impacts on productivity in establishing equitable actions for career progress for scientists during pandemics.

study participants did not consent to share any data publicly. The authors do not have the Institutional Review Board's approval to share the collected data publicly." Inquiries for data access or ethical issues may be asked to the Yale University Human Subjects Committee (Tel: 203-785-4688; email: human.subjects@yale.edu) or the Yale Human Research Protection Program (mail address: P.O. Box 208327, New Haven, CT 06520-8327; Tel: 203-785-4688).

**Funding:** This study was funded by Yale Women Faculty Forum (WFF) 2021-2022 Seed Grant awarded to Prof. Michelle L. Bell [available at https://wff.yale.edu/what-were-doing/research/seed-grants]. The funder had no role in study design, data collection and analysis, decision to publish, or preparation of the manuscript.

**Competing interests:** The authors have declared that no competing interests exist.

## Introduction

The COVID-19 pandemic that received international attention in (January 2020) was declared a public health emergency of international concern by the World Health Organization (WHO) in March 2020 [1]. The global health crisis caused by COVID-19 has upended daily and work routines in general. Much of the work has encountered unexpected changes such as transition from conventional work environment to the remote work environment, increased work demands, and lack of institutional support, which have been associated with emotional stress from uncertainty and pressure of new workspaces [2]. Furthermore, the COVID-19 pandemic has widened gender inequalities across many domains such as work, income, domestic duties, childcare responsibilities, and health [3, 4]. Globally, female-dominated job sectors (e.g., hospitality or childcare) tended to be more negatively affected before other job sectors dominated by male gender such as construction [3]. Mothers with young children had work hours reduced four to five times more than fathers during the COVID-19 compared to the pre-pandemic periods resulting in 20 to 50 percent increase in the gender gap in work hours between men and women [5]. The estimates of women in poverty in 2021 was 118 for every 100 men in poverty globally, and this was expected to rise to 121 women in poverty for every 100 men in poverty due to the economy crisis caused by the COVID-19 pandemic [6]. While the rate of COVID-19 infection was twice as high for men as for women [3], female gender was more associated with greater risks of distress during the pandemic [7, 8]. The uncertainty and worsened gender inequity across various sectors due to the pandemic shed light on the need for more support and protection for employees, especially women or marginalized persons, and more understanding of the impacts of the pandemic on gender inequality in the workplace.

The COVID-19 pandemic has caused unprecedented challenges for work routines and working conditions in academic fields [9]. These challenges include transition to remote working and online teaching, shutdown or limited access for laboratories and field work, decreased/delayed funding, and reduced in opportunities to network with peers, mentors, and collaborators. Along with these changes, productivity of academics and scientists have been challenged in various aspects. Many studies have reported a decline in work hours especially in fields involving physical laboratory activities (i.e., animal experiments) [10]. Many faculty were unfamiliar with online teaching software prior to the pandemic, which increased workloads and time for preparing for online teaching leaving less time for other activities and increased work-related stress [11]. Although more research has been called for on the impacts of the COVID-19 pandemic on scientific productivity and consequent health effects [12], research remains limited and more evidence is required to understand the obstacles scientists face during the pandemics, including how the challenges on productivity and well-being are disproportionately affected across different groups in academia.

Academicians also face many of the challenges for gender equality that impacted those in other professions such as eldercare and childcare. Gender gaps have been shown for academic fields and science [5, 13], and COVID-19 has amplified these gender inequalities in academia [14, 15] that have been long discussed for decades. Well-known factors for the existing gender inequality are the unequal distribution of care and domestic labor between men and women [16]. The gender bias that women do not have time for collaboration due to other many duties including childcare may hinder female scientists from engaging in scientific roles [17]. Women also generally perform more 'soft service' such as mentoring and administrative work in academia compared to men [18, 19]. These disparities can also relate to women not being as networked or recognized as professional leaders in academic fields as men [15]. The stay-at-home order in many countries and states led to closure of about 90% of childcare facilities and schools by April 2020 and [13] affected the productivity among scientists especially for those

who have children, and especially for those who are the primary caretakers of children [20]. Recent studies found that women usually carried a heavier load for childcare and domestic work than men and mother's increased workload for childcare during the pandemic was significantly associated with a decline in working hours in academic fields [10, 21–23]. Studies suggested that COVID-19 would affect retention, promotion, or tenure for male scientists but more adversely for female scientists [13].

Research publication is an imperfect but important and useful measure of academic productivity to identify trends over time and compare groups of people or periods. Studies suggested unequal impact of COVID-19 pandemic on publication in various fields [24]. For example, women's representation was lower for first and last authorship positions for papers related to COVID-19 [25]. Submission rates in public health fields were higher overall during the pandemic compared with before; however, increases were higher for submissions from men compared with women (41.9% vs 10.9% for corresponding author) [18]. The relative growth of submission to pre-print archives (arXiv, bioRxiv) for 2020 compared to 2019 was lower for women's authorship than men [13]. Another study found that the number of pre-print articles in 2020–2021 decreased by 15.2% worldwide, and the reduction in authorships was similar between men and women in Europe and North America, whereas the reduction was about two times larger for women than men in the Far East [26]. Altered childcare demands emerged as a consequence of the COVID-19 pandemic for females scientists and was identified as a factor that compromised ability to submit scientific articles for women [27]. Another study found that male academics with children were the least affected groups during the pandemic for the ability to submit papers as planned, whereas women with children, especially people of color, were the most impacted groups [21]. However, using the publication of scientific articles as a measure of performance or productivity has limitations as publication does not fully reflect common side-lining of other important pillars of academic labor such as teaching, mentorship, public outreach, community involvement, fund raising, reviewing papers, editorial duties, and other academic services, as well as ongoing and planned papers. Therefore, understanding the impact of COVID-19 on academicians for factors in addition to publication as a metric of research is needed to establish effective countermeasures to facility scientists' productivity and career progress in relation to the pandemic.

The mental health effects from the COVID-19 pandemic and related distancing measures would affect the ability to focus on work and academic productivity as well. Recent studies found evidence of high frequencies of mental health problems among university students, faculty, and staff during the pandemic worldwide [9, 28–33]. These studies found that female gender, unemployment status, isolation from colleagues due to social distancing, poor supervision, and financial crisis were associated with adverse mental health status. As many studies on the impact of COVID-19 on high education focused on students, relatively less is known for the impact of COVID-19 on faculty and scholars, which warrants further studies [20]. Links have been established between poor mental health status with motivation and productivity of scientists [34]. However, there is still a knowledge gap for the mental health status of scientists during the pandemic and how the challenges scientists face for work and career during the pandemic are associated with their mental health status as well as productivity.

Several studies have proposed that academic institutions and funding agencies should carefully build COVID-19 strategies ensuring sustainable career development for women with children [35]. Feasible strategies for higher-education institutions include acknowledging and monitoring sex breakdowns in promotion and tenure processes, extending currently funded grant periods; extending the tenure clock; updating tenure and promotion standards; investing in high-quality childcare support; reallocating the workloads among research, teaching, and service; and prioritizing grant supports for early-career women [13, 35]. While a few studies

have reported cases of academic institutions implementing proactive actions for diminishing gender gaps in academia during pre-pandemic periods, there is a need for understanding the effectiveness and perceptions of implemented academic policies during the pandemic for productivity and career development.

In this survey-based research, we investigated whether there are differences in the perception of work productivity and the challenges of productivity during the pandemic by gender during the pandemic in Science, Technology, Engineering, Mathematics, and Medicine (STEMM) fields. We also surveyed the implementation of institutional policies for COVID-19 and their impacts on the work productivity and mental health of the survey participants. This study aims to aid decision makers and leaders in academic fields by providing insights for implementing effective, rapid, and proactive responses for academic and scientific activities during this health crisis.

## Materials and methods

### Setting and participants

In this survey research, we targeted scientists working in research and/or an educational institution (e.g., university/college), government agency, industry, and other institution for Science, Technology, Engineering, and Mathematics (STEM), Medicine, Public Health, or other areas of science/engineering (hereafter referred as "STEMM fields"). Data were obtained through an online anonymous survey that were available in six languages: English, Portuguese, Spanish, Mandarin Chinese, Korean, and Japanese. The survey was sent via email to 2,314,691 individuals, who had published one or more research articles registered at Scopus and PubMed during 2017–2021, and the survey was open for October 5, 2021–December 31, 2021. To identify potential survey participants, we searched the published research articles in PubMed and Scopus for the academic journals related to the STEMM fields and exported the corresponding authors' email address without any other identifiable information. This approach was based on published research using this method [36] The search was applied to the most recent 5 years (2017–2021) to target scientists who are likely to be most active in the sciences. We also recruited participants by advertising our survey questionnaire on social media (Facebook, Instagram), which directed persons who clicked the link on social media to our questionnaire. The first page of the questionnaire was the online consent form and the remaining pages of the survey were only shown to, and only continued collecting responses from, participants who signed and submitted the written informed consent form. Verbal consent forms were not obtained from the participants as this survey was anonymous (no collection for personal information), there were no direct contacts between the participants and authors, and the survey was conducted entirely online. The following three screening questions, in the next pages of the survey, identified target participants who met all of the following conditions: (1) adults age ≥18 years old, (2) not current students, and (3) employed by institutions in the STEMM fields. Our survey research and consent procedure were reviewed and approved by the Yale University Institutional Review Board (IRB) Human Subjects Committee (Protocol ID #2000031343).

### Survey data

Our questionnaire had six sections focusing on (1) basic participant information, (2) work-related information, (3) responsibility for caregiving responsibility, (4) perception in work productivity, (5) factors affecting productivity at work, and (6) health status. The section of basic information asked about field of study, the country of residence, sex/gender, race/ethnicity, marital status, employment status, the number of years working in the current job, type of institution, highest education, and the number of years since obtaining the highest degree.

Work-related questions collected data for the percent of time devoted to research, teaching, service/administration, and other academic labors, academic position (postdoc, researcher, instructor, professor, leadership roles), tenure status (if applicable), the relevance of work to 'wet' science or living organisms. In the fourth section of the survey, we asked about the number of children <18 years living at home with the participants, age of the youngest child living at home, primary caregiver for children before and during the pandemic (e.g., participant, partner or spouse, other family member, paid childcare provider, childcare facilities), and changes in the amount of time spent on childcare, eldercare, and/or domestic work. We faced a challenge regarding the formation of the questions for 'sex' or 'gender' as the concepts of genders are not consistent across cultures, countries, or time [37–39], hindering our ability to analyze gender in this study, which collected data in six languages from 132 countries (S1 Table). We asked participants to identify their sex/gender as: male, female, transgender, gender non-conforming, or other with the opportunity to write-in text for 'other'. While 'sex' or 'gender' are not interchangeable terms, the survey was designed to be easily interpretable across a wide range of cultures and countries. We recognize that gender and sex are separate concepts, that many other genders exist, and that non-male/female persons exist everywhere [40–42]. Analysis focuses on 'male' and 'female' as the percentage of participants self-identified as 'other sex' was too small (1.1%) for statistical analyses.

In the fourth section regarding productivity, we asked "what is the approximate change in the number of work hours per week during the COVID-19 compared to the pre-pandemic period" with answers selected among 'significantly decreased', 'slightly decreased', 'neither decreased nor increased', 'slightly increased', and 'significantly increased'. Then, we asked how much participants agreed with each of the following statements, which are the perception of productivity: (1) "I am worried/concerned that the number and quality of my research papers and proposals has or will decrease due to the COVID-19 pandemic", (2) "I am worried/concerned that the COVID-19 pandemic has or will decrease my ability to mentor students and others", (3) "I am worried/concerned that the quality of my teaching has or will decrease due to the COVID-19 pandemic", (4) "I am worried/concerned that COVID-19 pandemic will have long-term effects on my promotion and career", and (5) "I am satisfied with my current career progress." For these five questions about perception in productivity, participants were asked to choose one answer among 'strongly agree', 'agree', 'neither agree nor disagree', 'strongly disagree', and 'not applicable'.

The questionnaire asked to each participant whether they were experiencing COVID-19 illness of family members of self, suspended classes and transition to remote teaching, restricted access to office, laboratories, or campus, decreased or delayed funding for research, delayed or halted fieldwork on other research, challenges in recruitment of research participants, elimination or restructuring of department or institution, considering retirement or forced retirement, increased childcare or eldercare at home, increased family care, poor home workspace or work conditions at home, and/or work travel restrictions.

We asked participants about policies for COVID-19 implemented by their institutions. The question asked the participants to check all answers that applied to them among the following answers or 'not sure': 'extending tenure clock', 'extending contract for the position', 'providing/facilitating contract for the position', 'providing/facilitating remote working environment', 'transitioning of leaning environment to online classes', 'creating online learning platforms (e.g., software)', 'providing rapid actions for COVID-19 research (e.g., expedited IRB process, etc.)', 'allowing physical access to resources in labs (experiment equipment)', 'adding support for mental health and well-being', 'adding support for work-life balance and family', and 'scheduling meetings to avoid early or late hours'. Then, we asked whether each of the selected policies helped the participant's productivity and career during the pandemic ('not likely',

'somewhat likely', 'very likely', 'not applicable').The mental health status was assessed using the Depression, Anxiety and Stress Scale—21 Items (DASS-21) [43], which is a reliable and valid standardized measure in assessing mental health status [7, 44]. It is a combination of 21 self-report questions designed to measure the emotional states of depression, anxiety, and stress. Each of the three scales contains seven items, divided into subscales with similar content. Survey participants rated the extent to which they experienced each subscale over the past week among the score of 0 to 3. The rating scale was 'did not apply to me at all' for score 0, 'applied to me to some degree, or some of the time' for score 1, 'applied to me to a considerable degree or a good part of time' for score 2, and 'applied to me very much or most of the time' for score 3. Scores for each scale were calculated by summing the scores for the relevant subscales. The severity of depression was divided into normal (score 0–9), mild (10–13), moderate (14–20), severe (21–27), and extremely severe ($\geq$28). The total anxiety score was divided into normal (0–6), mild (7–9), moderate (10–14), severe (15–19), and extremely severe ($\geq$20). The total stress score was divided in to normal (0–14), mild (15–18), moderate (19–25), severe (26–33), and extremely severe ($\geq$34).

## Statistical analysis

Statistical analyses were applied to assess the following: (1) changes in the number of work hours and perception in work productivity comparing the pandemic to the pre-pandemic period by sex and status as a parent, (2), associations of productivity during the pandemic and mental health status with individual-level factors including sex and status as a parent of one or more children age <18 years living at home, (3) and helpfulness of the COVID-19 actions taken at work. The analyses were limited to female and male groups.

The percentages of responses for the categories of changes in work hours (i.e., significantly decreased, slightly decreased, no change, slightly increased, significantly increased) by sex (female, male) and status as a parent (yes/no) for one or more children <18 years were compared using bar charts stratified by type of work (i.e., research, fundraising, teaching, administrative work or service), percentages for changed work hours with the 7 categories (decrease of 25 hours/week or more, decrease of 10 to 24 hours/week, decrease of 1 to 9 hours/week, 0 hours (i.e., no change), increase of 1 to 9 hours/week, increase of 10 to 24 hours/week, increase of 25 hours/week or more). The chi-square test was used to examine the statistical differences in the changed work hours by sex and status as a parent. The percentages of participants experiencing the obstacles of work and personal difficulties due to the pandemic were compared for females versus males and parents versus non-parents using a chi-square test.

To assess the factors associated with perceptions of productivity, we performed linear multivariable regression analyses. These models were separately applied to each statement for perception in productivity: (1) number of research papers and proposals, (2) teaching quality, (3) ability to mentor, (4) long-term effects of COVID-19 on career, and (5) and satisfaction for current career progress. The outcome variables for these perception statements were based on the Likert scale scores of 1 for 'strongly agree' to 5 for 'strongly disagree'. The explanatory variables included in the models were sex (female, male); early-career status (yes, no), status as a parent of children <18 years (yes, no); working on lab experiments, living organisms, or bench-science work (yes, no); changes in work hours during the pandemic (significantly decreased, slightly decreased, no change, slightly increased, significantly increased); satisfaction for workspace at home (numeric variable, scale of 0 to 10); satisfaction for work condition at home (numeric variable, scale of 0 to 10); suspended class (yes, no); restricted access to workspace and office (yes, no); decreased funding for research (yes, no); delayed or halted field work or other research (yes, no); challenges in recruitment of research participants (yes, no);

elimination or restructuring of department (yes, no); and restricted work travels (yes, no). Early-career status was assigned to job positions of postdocs, assistant professors, and those who obtained their highest degree within the last four years [45]. These explanatory variables were chosen based on the previous literature [9, 28–32].

The linear multivariable regression models were applied to the summed DASS scores for depression, anxiety, and stress, separately. The models included the following explanatory variables: employment status (currently employed, not employed), sex (female, male), marital status (single, married, divorced/widowed/separated, living with a partner), age (19–29, 30–59, 60+ years), loss of family due to COVID-19 (yes, no, prefer not to say), working with COVID-19 confirmed patients or in place with high contact with COVID-19 patients (yes, no, prefer not to say), and changes in the number of work hours (significantly decreased, slightly decreased, no change, slightly increased, significantly increased). A binary variable (i.e., yes/no) for each of the following information was also included in the models: early-career status, working in wet-science fields, status as a parent, diagnosis of mental health problems in last 12 months, experiencing loss of job, loss of job of spouse/partner, salary cut or paycheck delay, experiencing financial difficulties, reduced contract renewal or other changes in job security, considering early retirement or being forced to retire, restricted access to workspaces, decreased or delayed funding for research, delayed research work, challenges in recruitment of research participants, elimination or restructuring of department of institution, poor workspace or work condition at home, and restriction on work travels.

We calculated the percentages of votes among the participants on the Likert scales (1: not likely, 2: somewhat likely, 3: very likely), which indicate the degree to which each COVID-19 action is perceived to be helpful for productivity. All analyses were analyzed using Rstudio Version 1.4.1717 [46].

## Results

### Characteristics of participants

Of the total sample ($n$ = 4,494), 2,660 (59.2%) participants self-identified their sex as male and 40.8% as female (Table 1). The region with the highest number of survey respondents was Europe ($n$ = 1,575; 35.0%) followed by North America ($n$ = 1,302; 29.3%) and Asia ($n$ = 951; 21.4%). The largest group of participants was in medicine/public health/health science ($n$ = 1,701; 37.9%) followed by biology ($n$ = 1,014; 22.6%). Of the total participants, 75% were employed by universities and 54.6% were assistant professors, associate professors, or professors. Among the total participants, 32.4% were researchers of which 9% were postdoctoral researchers. A total of 41.7% of participants reported that they have one or more children <18 years living at home. Most participants (87.7%) were in the age group 30–64 years ($n$ = 3,945; 87.7%).

### Changes in work hours and academic labors

Changes in the number of work hours, comparing the pandemic to pre-pandemic periods, showed that a substantial portion of scholars had increased work hours (45.9%) but a substantial portion were working less (21.7%). These data by sex and status as a parent of children age <18 years are shown in Fig 1. Women showed higher percentages of increased work hours than men (53.2% versus 41.0%). Within the same sex, non-parents showed higher percentages of increases in the number of work hours than non-parents. The percentage for decreased working hours (significant and slight decreases) was the highest among the male participants who were also parents (28.9%).

**Table 1. Characteristics of the survey participants.**

| Group | n (%) |
|---|---|
| Total | 4,494 (100) |
| Sex (self-reported) | |
| Male | 2,660 (59.2) |
| Female | 1,834 (40.8) |
| Regions | |
| Africa | 162 (3.6) |
| Asia | 951 (21.4) |
| Caribbean | 6 (0.1) |
| Europe | 1,575 (35.0) |
| North America | 1,302 (29.3) |
| South America | 286 (6.4) |
| Oceania | 173 (3.9) |
| Other | 11 (0.2) |
| Field (multiple selections permitted) | |
| Agriculture/natural science | 344 (7.7) |
| Astronomy/astrophysics | 62 (1.4) |
| Biology | 1,014 (22.6) |
| Chemistry | 371 (8.3) |
| Computer science | 289 (6.4) |
| Environment/earth science | 461 (10.3) |
| Engineering | 591 (13.2) |
| Geology | 87 (1.9) |
| Interdisciplinary research | 434 (9.7) |
| Mathematics | 264 (5.9) |
| Medicine/public health/health science | 1,701 (37.9) |
| Physics | 300 (6.7) |
| Zoology/Animal science | 144 (3.2) |
| Other | 468 (10.4) |
| Institution (for those currently employed) | |
| University | 3,231 (75.0) |
| Government agency | 491 (11.4) |
| Industry | 188 (4.4) |
| Other | 399 (9.3) |
| Position | |
| Postdoctoral researcher | 405 (9.0) |
| Researcher | 1,050 (23.4) |
| Instructor | 183 (4.1) |
| Assistant professor | 597 (13.3) |
| Associate professor | 828 (18.4) |
| Professor | 1,029 (22.9) |
| Dean/provost/other leadership position | 152 (3.4) |
| Other | 246 (5.5) |
| Early-career status | |
| Early career | 1,004 (22.3) |
| Non-early career | 3,488 (77.6) |
| Tenure status | |
| Tenured | 1,529 (52.4) |

(*Continued*)

**Table 1.** (Continued)

| Group | *n* (%) |
|---|---|
| Tenure-track | 467 (16.0) |
| Not tenure track | 923 (31.6) |
| Status as a parent of children age <18 years | |
| Yes | 1,871 (41.7) |
| No | 2,618 (58.3) |
| Age (years) | |
| 18–19 | 1 (0.0) |
| 20–24 | 8 (0.2) |
| 25–29 | 165 (3.7) |
| 30–34 | 496 (11.0) |
| 35–39 | 752 (16.7) |
| 40–44 | 772 (17.2) |
| 45–49 | 616 (13.7) |
| 50–54 | 540 (12.0) |
| 55–59 | 415 (9.2) |
| 60–64 | 354 (7.9) |
| 65–69 | 199 (4.4) |
| 70–74 | 102 (2.3) |
| 75+ | 74 (1.6) |
| Highest degree | |
| High school | 27 (0.6) |
| Undergraduate | 39 (0.9) |
| Master's degree | 371 (8.3) |
| PhD | 3,670 (81.7) |
| Clinical degree | 591 (13.2) |
| Other | 120 (2.7) |

We compared the changes in the number of work hours for different types of work by sex and status as a parent (Fig 2). For the activities of fundraising, teaching, and administrative work/service, more participants had increased hours (59.9%) than decreased hours (29.3%). The percentage of participants with increased time for teaching (p-value = 0.001) and administrative work/service (p-value < 0.001) was significantly higher for women than men. The participants as a parent for one or more children <18 years showed significantly lower frequency of increased work hours devoted to research (p-value < 0.001) and fundraising (p-value < 0.001) than the participants without children.

We asked the survey participants about their experiences of obstacles of work and home-life balance during the pandemic (Table 2). The obstacle of work faced by the most participants was 'restricted access to campus, office, laboratories, field work, or other facilities' (n = 3,677), experienced by 81.8% of the participants. Women had significantly higher percentages for 6 types of work obstacles than men, based on the chi-squared test with a significance level of 0.05. For personal difficulties, responses were not statistically different by sex. Significantly more parents (children age <18 years) than non-parents experienced five of the ten work obstacles: 'suspended class / transitioning to remote teaching', 'decreased or delayed funding for research', 'delayed or halted field work or other research work', 'poor home workspace or work conditions at home', and 'considering retirement / forced retirement'. The percentages of experiencing 'salary cut or paycheck delay' (13.7%) and 'financial difficulties' (12.0%)

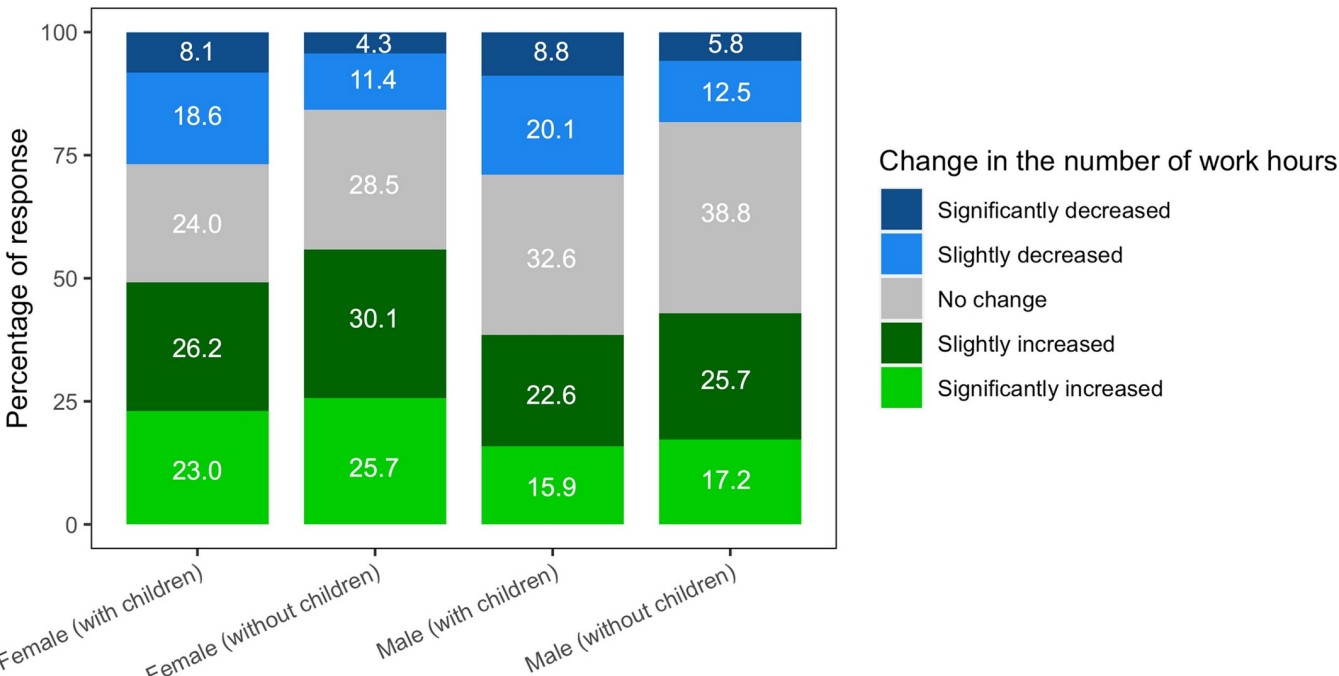

**Fig 1. Changes in the number of work hours during the pandemic by sex and status as a parent (children <18 years).**

among the participants who were parents of children <18 years were higher than non-parents and the percentages among the total participants (11.5% for each). Female participants faced more childcare/eldercare responsibilities than men (37.1% versus 32.8%; p-value = 0.004).

## Perception of productivity

Regression analysis was applied to a 5-level Likert scale (1: strongly agree, 5: strongly disagree) for perception of productivity during the COVID-19 (Table 3). Results showed that concerns for decreases in the number of research papers and proposals, and for long-term career progress due to the COVID-19 pandemic were significantly higher in women than men. Early-career status was significantly associated with increased concerns on the effects of the pandemic on research and career progress. Higher satisfaction in the workspace and work conditions at home were associated with more concerns for the effects of COVID-19 on productivity. Most work-related challenges were significantly associated with concerns on productivity across research, mentoring, teaching, and career progress. Especially, restricted access to workplace was significantly associated with higher all the concern for productivity and career.

In summary, female scientists were more likely to be concerned about the negative effects of the COVID-19 pandemic on their number of research papers and proposals and their long-term career progress and less likely to be satisfied with their current career progress than male scientists were. On the contrary, female scientists were less worried about the negative effects of the pandemic on their teaching quality and ability to mentor students and others. Status as a parent of children age <18 years was not associated with higher or lower concerns on productivity and career. Higher scores for satisfaction for workspace and work conditions at home were significantly associated with decreased concerns for the number of research papers and proposals, teaching quality, ability to mentor, and long-term career development. Lower Likert

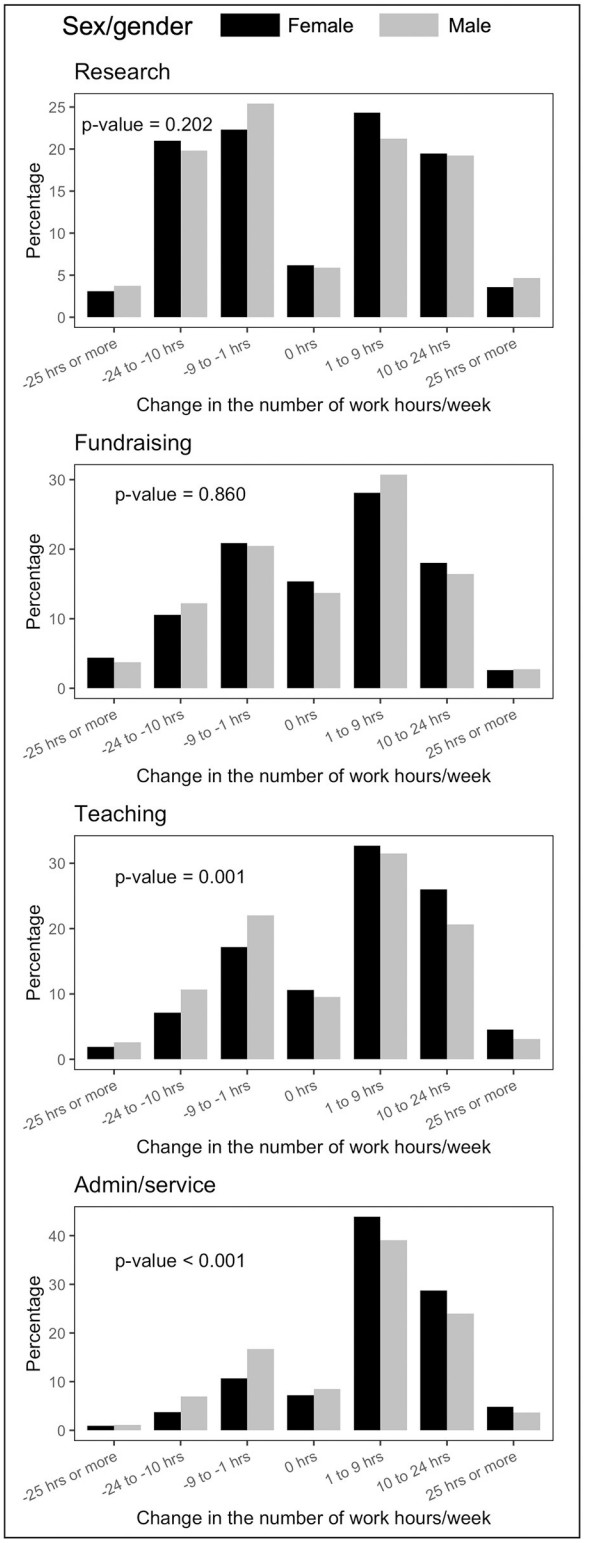
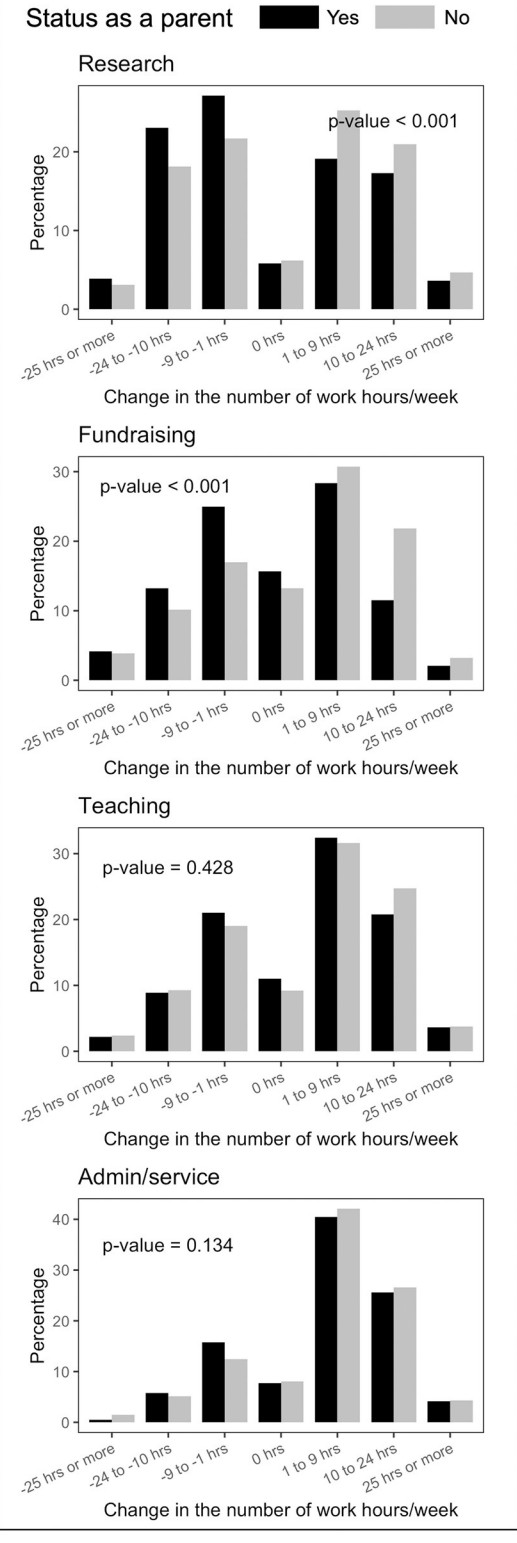

**Fig 2. Changes in the number of work hours by type of academic labors, stratified by sex and status as a parent for children <18 years (n = 4,494).**

**Table 2. Percentage of respondents experiencing the obstacles of work and home-life situations due to the pandemic (*n* = 4,494).**

| | Total (*n* = 4,494) | Sex | | | Status as a parent* | | |
|---|---|---|---|---|---|---|---|
| | | Female (*n* = 1,834) | Male (*n* = 2,660) | p-value of X² test | Parent* (*n* = 1,871) | Non-parent (*n* = 2,618) | p-value of X² test |
| **Obstacles of work** | | | | | | | |
| Suspended class / transition to remote teaching | 3115 (69.3) | 1258 (68.6) | 1857 (69.8) | 0.402 | 1343 (71.8) | 1771 (67.6) | 0.003 |
| Restricted access to campus, office, laboratories, field work, or other facilities | 3677 (81.8) | 1541 (84.0) | 2136 (80.3) | 0.002 | 1541 (82.4) | 2134 (81.5) | 0.491 |
| Decreased / delayed funding for research | 1540 (34.3) | 669 (36.5) | 871 (32.7) | 0.010 | 679 (36.3) | 859 (32.8) | 0.017 |
| Delayed or halted field work or other research work | 2426 (54.0) | 1074 (58.6) | 1352 (50.8) | <0.001 | 1057 (56.5) | 1367 (52.2) | 0.005 |
| Challenges in recruitment of research participants | 1906 (42.4) | 891 (48.6) | 1015 (38.2) | <0.001 | 810 (43.3) | 1095 (41.8) | 0.342 |
| Elimination or restructuring of department or institution | 570 (12.7) | 261 (14.2) | 309 (11.6) | 0.011 | 240 (12.8) | 330 (12.6) | 0.861 |
| Poor home workspace or work conditions at home | 1674 (37.2) | 781 (42.6) | 893 (33.6) | <0.001 | 829 (44.3) | 845 (32.3) | <0.001 |
| Work travel restrictions | 33535 (78.7) | 1444 (78.7) | 2091 (78.6) | 0.949 | 1494 (79.9) | 2039 (77.9) | 0.121 |
| Considering retirement / forced retirement | 219 (4.9) | 97 (5.3) | 122 (4.6) | 0.315 | 51 (2.7) | 167 (6.4) | <0.001 |
| Reduced or altered contract renewal or other change in job security | 473 (10.5) | 192 (10.5) | 281 (10.6) | 0.958 | 185 (9.9) | 287 (11.0) | 0.268 |
| **Obstacles for home-life situations** | | | | | | | |
| Loss of job | 77 (1.7) | 32 (1.7) | 45 (1.7) | 0.986 | 25 (1.3) | 52 (2.0) | 0.124 |
| Loss of partner's job | 164 (3.6) | 76 (4.1) | 88 (3.3) | 0.165 | 86 (4.6) | 78 (3.0) | 0.006 |
| Salary cut or paycheck delay | 516 (11.5) | 204 (11.1) | 312 (11.7) | 0.563 | 257 (13.7) | 257 (9.8) | <0.001 |
| Financial difficulties (for self or family) | 516 (11.5) | 215 (11.7) | 299 (11.2) | 0.652 | 225 (12.0) | 288 (11.0) | 0.309 |
| Increased childcare/eldercare demands | 1553 (34.6) | 680 (37.1) | 873 (32.8) | 0.004 | 1222 (65.3) | 330 (12.6) | <0.001 |
| Increased demands for domestic work | 2507 (55.8) | 1067 (58.2) | 1440 (54.1) | 0.008 | 1217 (65.0) | 1285 (49.1) | <0.001 |

Note. Multiple choices were allowed among the answers.

*: Parents of one or more children <18 years living together at home.

scale scores for concerns about long-term career development (i.e., higher concerns) were associated with restricted access to workplace, decreased funding for research, elimination or restructuring of department, and restriction of work travel.

## Institutional actions for COVID-19 and perception of productivity

The percentages of participants who responded that their institution had implemented COVID-19 actions during the pandemic by region are shown in Table 4. Results showed that transitioning to online learning occurred in all regions. Extension of the tenure clock was reported by less than 20% of participants in all regions except for North America, which was reported by 40.3% of participants. Extension of contract was highest in Africa (22.8%) followed by South America (19.8%). Providing options for remote working was implemented for less than 20% of participants in all regions. Creating online learning platforms was the most common institutional action, reported by a high percentage of participants in Caribbean countries (83.3%) and regions categorized as 'other' (81.8%), followed by North America (66.5%) and Asia (62.8%). The percentage of responses for providing mental health care was lowest in Asian countries (28.9%), while the percentage was higher than 40% elsewhere. Less than 20% of participants reported that their institution implemented actions to address work-life balance.

**Table 3. Results of regression analysis on the Likert-scale concerns and satisfactions for productivity during the pandemic (1: strongly agree, 2: agree, 3: neither agree nor disagree, 4: disagree, 5: strongly disagree; *N* = 3,289).**

| Variable | Beta (95% CI) | | | | |
|---|---|---|---|---|---|
| | *"I am worried that the number of my research papers and proposals has or will decrease due to the pandemic."* | *"I am worried that the pandemic has or will decrease my ability to mentor students and others."* | *"I am worried. that my teaching quality has or will decrease due to the pandemic."* | *"I am worried that the pandemic will have long-term effects on my promotion and career."* | *"I am satisfied with my current career progress."* |
| Sex | | | | | |
| Male | Reference | Reference | Reference | Reference | Reference |
| Female | -0.14 (-0.21, -0.07)* | 0.09 (0.01, 0.16)* | 0.13 (0.06, 0.20)* | -0.15 (-0.23, -0.08)* | 0.15 (0.08, 0.22)* |
| Early-career status | | | | | |
| No | Reference | Reference | Reference | Reference | Reference |
| Yes | -0.12 (-0.21, -0.03)* | -0.02 (-0.10, 0.06) | 0.01 (-0.08, 0.10) | -0.47 (-0.55, -0.38)* | 0.23 (0.15, 0.31)* |
| Parent of children <18 years | | | | | |
| No | Reference | Reference | Reference | Reference | Reference |
| Yes | -0.03 (-0.11, 0.04) | 0.01 (-0.06, 0.08) | 0.08 (0.01, 0.15)* | -0.05 (-0.13, 0.02) | 0.05 (-0.02, 0.12) |
| Working in the fileds involving lab experiments, bench science work, wet-science, and living organisms | | | | | |
| No | Reference | Reference | Reference | Reference | Reference |
| Yes | -0.36 (-0.44, -0.29)* | -0.11 (-0.18, -0.04)* | -0.02 (-0.09, 0.05) | -0.16 (-0.23, -0.08)* | 0.00 (-0.07, 0.07) |
| Changes in work hours during the pandemic | | | | | |
| Significantly decreased | Reference | Reference | Reference | Reference | Reference |
| Slightly decreased | 0.23 (0.06, 0.40)* | -0.23 (-0.33, -0.13)* | 0.06 (-0.11, 0.23) | 0.31 (0.15, 0.48)* | -0.26 (-0.41, -0.11)* |
| No change | 0.71 (0.55, 0.87)* | -0.15 (-0.23, -0.07)* | 0.22 (0.06, 0.38) | 0.72 (0.56, 0.87)* | -0.46 (-0.6, -0.32)* |
| Slightly increased | 0.51 (0.36, 0.67)* | -0.07 (-0.14, 0.01) | 0.07 (-0.09, 0.23) | 0.46 (0.3, 0.61)* | -0.37 (-0.52, -0.23)* |
| Significantly increased | 0.41 (0.25, 0.58)* | -0.17 (-0.24, -0.09)* | 0.11 (-0.05, 0.27) | 0.41 (0.25, 0.57)* | -0.25 (-0.4, -0.11)* |
| Suspended class | | | | | |
| No | Reference | Reference | Reference | Reference | Reference |
| Yes | -0.06 (-0.14, 0.02)* | -0.19 (-0.29, -0.08)* | -0.11 (-0.2, -0.02)* | 0.07 (-0.01, 0.15) | -0.02 (-0.10, 0.05) |
| Restricted access to workspace/office | | | | | |
| No | Reference | Reference | Reference | Reference | Reference |
| Yes | -0.19 (-0.3, -0.09)* | -0.05 (-0.14, 0.04) | -0.16 (-0.27, -0.06)* | -0.10 (-0.20, 0.00)* | 0.11 (0.02, 0.20)* |
| Decreased funding for research | | | | | |
| No | Reference | Reference | Reference | Reference | Reference |
| Yes | -0.42 (-0.5, -0.34)* | -0.07 (-0.15, 0.01) | -0.14 (-0.22, -0.06)* | -0.38 (-0.46, -0.30)* | 0.21 (0.13, 0.28)* |
| Delayed or halted field work or other research | | | | | |
| No | Reference | Reference | Reference | Reference | Reference |
| Yes | -0.08 (-0.16, 0.00)* | -0.23 (-0.33, -0.13)* | 0.00 (-0.07, 0.08) | -0.02 (-0.10, 0.06) | 0.04 (-0.03, 0.12) |
| Challenge in recruitment of research participants | | | | | |
| No | Reference | Reference | Reference | Reference | Reference |
| Yes | -0.07 (-0.15, 0.00)* | -0.15 (-0.23, -0.07)* | -0.11 (-0.19, -0.04)* | -0.09 (-0.17, -0.02)* | -0.01 (-0.08, 0.06) |

*(Continued)*

**Table 3.** (Continued)

| Variable | Beta (95% CI) | | | | |
|---|---|---|---|---|---|
| | *"I am worried that the number of my research papers and proposals has or will decrease due to the pandemic."* | *"I am worried that the pandemic has or will decrease my ability to mentor students and others."* | *"I am worried. that my teaching quality has or will decrease due to the pandemic."* | *"I am worried that the pandemic will have long-term effects on my promotion and career."* | *"I am satisfied with my current career progress."* |
| Elimination or restructuring of department | | | | | |
| No | Reference | Reference | Reference | Reference | Reference |
| Yes | -0.06 (-0.17, 0.05) | -0.07 (-0.14, 0.01) | -0.15 (-0.25, -0.04)* | -0.25 (-0.36, -0.14)* | 0.14 (0.04, 0.24)* |
| Restriction on work travel | | | | | |
| No | Reference | Reference | Reference | Reference | Reference |
| Yes | -0.02 (-0.12, 0.07) | -0.17 (-0.24, -0.09)* | -0.09 (-0.18, 0.00)* | 0.07 (-0.02, 0.16) | -0.02 (-0.10, 0.07) |
| Satisfaction for workspace at home† | 0.05 (0.03, 0.06)* | 0.03 (0.01, 0.05)* | 0.03 (0.01, 0.05)* | 0.04 (0.02, 0.05)* | -0.03 (-0.04, -0.01)* |
| Satisfaction for work condition at home† | 0.08 (0.06, 0.10)* | 0.06 (0.04, 0.08)* | 0.06 (0.04, 0.08)* | 0.09 (0.07, 0.11)* | -0.09 (-0.11, -0.07)* |

Notes

*: Significant at a significance level of 0.05. Participants with missing data were omitted.

†: Scale of 0 ('not satisfied at all') to 10 ('very satisfied').

We also asked participants their view on the effectiveness of these institutional actions (Table 5). More than half of the respondents agreed that extending contract of position and providing options during the pandemic for remote work were 'very likely' helpful for their productivity and career. About 45% of the respondents agreed that the COVID-19 actions including creating online learning platforms, providing rapid actions for COVID-19-related research, allowing limited physical access to workplaces, providing work-life balance care, and arranging meeting hours flexibly were 'very likely' helpful for their productivity and career. Transitioning to online learning was reported to be 'somewhat likely' helpful for 40% of the respondents. About 45.5% of respondents responded that the COVID-19 action to extend tenure clock was 'not likely' helpful for productivity and career.

**Table 4. Percentage of participants who responded that their institution had taken COVID-19 actions during the pandemic, by region (n = 4,494).**

| Institutional action | Overall | Region | | | | | | | |
|---|---|---|---|---|---|---|---|---|---|
| | | Africa | Asia | Europe | North America | South America | Caribbean regions | Oceania | Other |
| Extending tenure clock | 15.9 | 8.0 | 5.6 | 5.5 | 40.3 | 10.1 | 0.0 | 0.0 | 18.2 |
| Extending contract of position | 8.1 | 22.8 | 14.6 | 14.1 | 11.9 | 19.8 | 0.0 | 13.2 | 18.2 |
| Providing options for remote work | 50.7 | 8.6 | 5.6 | 12.4 | 6.0 | 5.6 | 0.0 | 6.3 | 0.0 |
| Transition to online learning | 61.1 | 45.7 | 46.2 | 50.8 | 57.7 | 34.0 | 50.0 | 56.9 | 36.4 |
| Creating online learning platforms | 44.1 | 54.9 | 62.8 | 57.7 | 66.5 | 45.1 | 83.3 | 73.6 | 81.8 |
| Providing rapid actions for COVID-19 related research | 17.2 | 61.7 | 62.6 | 42.9 | 29.9 | 48.3 | 33.3 | 32.8 | 54.5 |
| Allowing limited physical access to workplace | 44.0 | 21.0 | 15.0 | 13.6 | 22.5 | 17.7 | 0.0 | 14.4 | 18.2 |
| Providing mental health care programs | 27.4 | 43.8 | 28.9 | 45.5 | 51.4 | 46.9 | 50.0 | 54.6 | 54.5 |
| Providing work-life balance care | 13.0 | 16.0 | 19.2 | 20.4 | 40.2 | 23.6 | 66.7 | 51.1 | 45.5 |
| Arranging meeting hours flexibly | 14.3 | 10.5 | 10.8 | 11.3 | 16.1 | 10.4 | 16.7 | 20.1 | 9.1 |

Notes. Dark orange: 0–19, moderate orange: 20–39, yellow: 40–59, moderate blue: 60–79, dark blue: 80–100.

**Table 5. Percentage of participants who responded that the COVID-19 actions taken at work were helpful for keeping productivity and career (n = 4,494).**

| Action | Likert scale | | |
|---|---|---|---|
| | **Not likely** | **Somewhat likely** | **Very likely** |
| Extending tenure clock | 45.5 | 33.2 | 21.2 |
| Extending contract of position | 17.4 | 29.4 | 53.2 |
| Providing options for remote work | 11.2 | 38.1 | 50.7 |
| Transition to online learning | 28.0 | 40.0 | 32.5 |
| Creating online learning platforms | 16.0 | 39.7 | 44.3 |
| Providing rapid actions for COVID-19 related research | 20.6 | 33.1 | 46.3 |
| Allowing limited physical access to workplace | 20.0 | 35.6 | 44.4 |
| Providing mental health care programs | 30.9 | 39.4 | 29.7 |
| Providing work-life balance care | 20.8 | 32.2 | 47.0 |
| Arranging meeting hours flexibly | 15.0 | 40.0 | 45.0 |

Note. White cell: 0–19%, light grey cell: 20–39%, light blue cell: 40–59%.

## Mental health status

The severity of depression, anxiety, and stress were categorized as normal, mild, moderate, severe, and extremely severe (Table 6). Most participants scored in the normal category for all three indicators of mental health, although 21.2%, 15.5%, and 13.0% of participants had moderate to extremely severe levels of depression, anxiety, and stress, respectively. Severe or extremely severe levels of depression, anxiety, and stress were found in 7.5%, 5.4%, and 3.9% of participants, respectively.

Regression analysis was applied to the DASS-21 scores (separately for scores for depression, anxiety, and stress) (S2 Table). Results indicated that the following variables were significantly associated with depression, anxiety, and stress: being single, older age, loss of family members due to COVID-19, diagnosis of mental health problems in last 12 months, decreased number of work hours, experiencing financial difficulties, reduced contract renewal or other changes in job security, considering early retirement or being forced to retire, delays in research work,

**Table 6. Participants scores for severity of mild, moderate, or severe depression, anxiety, and stress.**

| Group | n (%) |
|---|---|
| DASS-21 –severity of depression | |
| Normal | 2,736 (66.5) |
| Mild | 506 (12.3) |
| Moderate | 566 (13.7) |
| Severe– extremely severe | 309 (7.5) |
| DASS-21 – severity of anxiety | |
| Normal | 3,249 (78.9) |
| Mild | 232 (5.6) |
| Moderate | 414 (10.1) |
| Severe– extremely severe | 221 (5.4) |
| DASS-21 – severity of stress | |
| Normal | 2,415 (59.1) |
| Mild | 1142 (27.9) |
| Moderate | 374 (9.1) |
| Severe– extremely severe | 159 (3.9) |

poor workspace or work condition at home, and increased demands for domestic work. Anxiety was also more likely for those working in wet-science fields, those who lost their job, and those whose spouse/partner lost their job. Stress was also higher in women and those with restricted access to campus or other work facilities. These results were generally robust across different regions (S3–S8 Tables).

## Discussion

The COVID-19 pandemic has disrupted academic labor and productivity across the world disproportionately affecting female scientists [16]. In analyzing gender inequities of academic labor during the COVID-19 pandemic, studies have usually measured productivity as the ability to work or as outputs [16]. Our study measured productivity based on time spent for academic work and perception of productivity. We found that time devoted to work increased for some participants but decreased for others during the pandemic compared to the pre-pandemic periods, indicating different effects of the pandemic across individuals. A previous survey study for scientists comparing the time devoted to research between 2019 and 2020 found that the difference in time for research was minor (average 7.1 hours decrease per week) [47]. The percentage of increased time for work was higher for women than men in our data. We examined the changes in work hours by sex and status as a parent stratifying the changes in work hours for research, fundraising, teaching, and administrative work. The results showed that female scientists faced more increased demands for teaching and administrative work compared to men. There was no significant difference for changes in time for research or fundraising by sex. Further, scientists with young children (e.g., age <18 years) were likely to have higher reduction in the number of work hours during the pandemic for both men and women. Given that women have unequally higher demands and loads for teaching and service in academic fields than men [48], these reductions in the time for research due to the COVID-19 pandemic have implications of unequal impacts of COVID-19 on research outputs and career achievements for women.

A few recent studies have found that some challenges due to the COVID-19 pandemic largely relate to a poor work environment, limited access to resources, new allocation of workloads, and lack of informal contact with colleagues for doctoral students and other researchers [49, 50]. A survey study conducted in March 2020 targeting university-based biochemists, biologists, and civil engineers found that university closure and disruption of lab work emerged as the most negative COVID-19 impacts for the majority of participants (88–93%) [23]. Similarly, we found that a large group of our survey participants in STEMM fields experienced emerging challenges for their academic labors due to the COVID-19 pandemic. We found that female participants faced more increased childcare/eldercare responsibilities than men during the pandemic, which is consistent with numerous previous study [23]. In our data, women and parents were more likely to experience negative changes for work during the pandemic such as decreased funding or delayed research activities. The majority of these challenges for work were further significantly associated with perception for decreased productivity for research, fundraising, and teaching. The differences for men and women for division of caregiving for children can lead to disproportionate opportunities for career progress (i.e., salary, access to funding) and may intensify beyond the pandemic.

Our results highlight the different experiences on productivity for men and women as well as status as a parent of children at home. We found that women were more likely to be concerned regarding the negative effects of the COVID-19 pandemic on their number of research papers and proposals and their long-term career progress, and less likely to be satisfied with their current career progress compared to male scientists. On the contrary, a previous survey

study conducted in May 2020 found little difference between men and women or between female parents and male parents for their perception of the negative impacts of COVID-19 on research productivity [20]. That study was based on the data of 1,003 respondents recruited from the International Studies Association, with 60% of participants from the US, and the comparison of productivity between men and women was based on the chi-square test. Thus, the differences in study results may be potentially due to different study participants, survey recruitment periods, measure of perception of productivity, and statistical analysis. Our results that perception of productivity by sex differed between research, teaching, and administrative work indicate that productivity cannot be simplified by publications or time devoted to work. The lower concerns for the impacts of COVID-19 on teaching and mentoring ability in women than men could be also associated with the practices in academic fields assigning women more service and teaching loads, affecting women's ability to obtain the same research achievements [48].

Some recent studies found adverse impacts of the COVID-19 pandemic on researchers and faculty's mental health [29, 31, 32], showing higher frequency of mental health problems. To the best of our knowledge, our study is the first to examine the associations between standardized mental health scores and aspects of work and life being impacted by the COVID-19 for STEMM scholars. Our results showed that many aspects of the changes for work and home-life situations were associated with high risks of depression, anxiety, and stress among the participants. We found that female participants were more stressed than male counterparts, whereas the risks of depression and anxiety did not differ between men and women. Status as a parent (children age <18 years) was not associated with increases in the DASS-21 scores, but increased demands for domestic work were associated with higher likelihood of feeling anxious and more stressed. The percentage of responses for increased demands for domestic work was significantly higher in women than men and parents of children <18 years than others, indicating higher risks of deteriorated mental health status among women and parents during the pandemic due to altered responsibilities for caregiving. Further research is needed to understand the associations between work-life imbalance and mental health status of scientists as the study results are scarce.

Inequities in opportunity for female principal investigators or faculty, especially during times of crisis, could have long-term impacts on career progress. Thus, it is important for funding bodies, institutions of higher education, research organizations, professional societies, and journals to consider the impacts of the pandemic on female scientists such as altered childcare demands [51]. Our results suggest the need of institutional policies to diminish differences by gender or sex and the impacts of COVID-19 for scientific work. We suggest that decision makers and leaders of academic and related scientific institutions should provide supports for female academics and those with families in higher education during and beyond pandemic times. We also emphasize the importance of the institutional policies to monitor the allocation of workloads for research, teaching, and service between female and male scientists, which have been suggested by previous studies as disparate [35]. In attempts to compensate for work challenges due to COVID-19, many institutions have offered tenure clock extensions [48]. Our results found that most participants did not consider this strategy to be effective. This may be due to many factors, such as the risk that extending tenure can prevent faculty from some benefits of tenure although it can be a stress relief measure for first-year faculty on their tenure track [48]. Our results found that about 45–50% of the participants agreed that remote working and allowing limited access to workplaces were very likely helpful for productivity. This may indicate that finding solutions to ensure flexible choices between remote work and commuting to work while minimizing the risk of disease spread in a timely manner can be important parts of a response strategy. In such process, the characteristics of work (i.e.,

need for access to laboratories) must be considered. About the half of the participants also responded that providing work-life balance support was very likely helpful for productivity. Childcare can be an important part of solutions, such as providing facilities and services to take care for children in the workplace, for scientists with very young children and for those conducting research activities that are difficult to perform remotely (e.g., field work).

Our study has several strengths. While most previous studies limited the analysis for sex or gender gaps for productivity during the pandemic an individual field or certain countries, we examined this issue using the survey data for various disciplines in the STEMM fields across various regions represented by six languages. We quantitatively examined the associations between factors of work and home-life situations during the COVID-19 pandemic, productivity, and mental health status. The mental health status was examined for 3 different outcomes (depression, anxiety, and stress) using a standardized questionnaire widely used in the studies of mental health.

A limitation of this research is that we did not have information how the COVID-19-related actions were changed, modified, or halted along with the waves of COVID-19 pandemic from our survey data as our research is a cross-sectional study. Further, we were unable to distinguish between national and local pandemic policies across the broad regions of the participants. Our analysis is limited to the STEMM fields only so does not represent scholars in other fields such as art and humanities. Participants who were more interested in the topic of gender and sex gaps and/or the pandemic may have been more likely to respond to our survey request, which may affect the results of our analysis through selection bias. Although we used published methods to identify participants, when we sent our email invitations for the survey to the searched corresponding email addresses from the publication database, a few recipients raised concerns that they had never consented to receive survey invitations, and their corresponding author information was posted for inquiries regarding their articles and not for other purposes although their email addresses were publicly available. Thus, the method used in our study to find potential participants may be inappropriate for application to other studies in the future, and we do not recommend this approach although it was approved by the IRB. Our assessment of depression, anxiety, and stress used published, validated measures, but cannot capture the full spectrum of mental health and wellbeing. While we examined differences in impacts and perceptions by several factors, there are many other subpopulations that are likely to be disproportionately affected, such as racial/ethnic minority and other underrepresented scientists such as those from low-income backgrounds [52–55], scholars who are providing elder care, and those who experienced COVID-19 or had family members or close friends experienced COVID-19. Lastly, we recognize that the distinction between sex and gender is critically important and that there are multiple genders [40–42, 56], we were challenged by how to address this issue in a context where study participants were from a range of cultures [37–39, 56–58]across 132 countries and six languages where these concepts differ considerably. We therefore used the term 'sex' to describe a variable in this study that likely represented sex for some participants and gender for others, and must acknowledge this critical limitation. The concepts of sex and gender are interpreted and defined in non-Western countries in different ways than they are in Western countries. Despite the differences in these concepts, the English words 'sex' and 'gender' are commonly used interchangeably in conversations in countries where there are no separate words for biological sex and gender but there is only one word referring them together. Even though our questionnaire asked the participants to self-report their 'identified sex or gender' among answers of 'female', 'male', 'transgender', 'gender nonconforming', and 'other', the impacts of COVID-19 on the academic fields may differ between biological sex and self-identified social genders, which warrant substantial future studies with

more detailed subgroups for sex and gender. Such work may need to use language that is specific to the culture of study participants.

In summary, for a study of 4,494 participants in six languages, we found that for many STEMM scientists work hours increased during the pandemic compared to pre-pandemic periods, while for many others work hours decreased, and we identified differences for the changed number of work hours for academic labors by sex and status as a parent of children <18 years. Many of the scientists who participated in our study experienced various obstacles for work and home-life situations due to the pandemic, with the most commonly identified barrier as restricted access to campus, laboratories, and other workspaces. These COVID-19-related obstacles were significantly associated with increased risks of depression, anxiety, and stress among the participants. Women were more likely than men to be concerned about the impact of the pandemic on their long-term career. Future studies should extend the discussion for institutional policies for the scientific workforce to cope with the gender and sex differences and academic challenges during a crisis and more broadly. Research should be also extended to other marginalized subgroups of race/ethnicity, the more complete and complex spectrum of genders, and rank of job positions of scientists, as well as scholars in other fields.

## Supporting information

**S1 Table. List of countries where the survey participants reside.**
(DOCX)

**S2 Table. Results of multivariate regression analysis for DASS-21 scores of depression, anxiety, and stress ($n$ = 3,893).**
(DOCX)

**S3 Table. Results of multivariate regression analysis for DASS-21 scores of depression, anxiety, and stress for the participants in Africa (n = 162).**
(DOCX)

**S4 Table. Results of multivariate regression analysis for DASS-21 scores of depression, anxiety, and stress for the participants in Asia (n = 951).**
(DOCX)

**S5 Table. Results of multivariate regression analysis for DASS-21 scores of depression, anxiety, and stress for the participants in Europe (n = 1,555).**
(DOCX)

**S6 Table. Results of multivariate regression analysis for DASS-21 scores of depression, anxiety, and stress for the participants in North America (n = 1,302).**
(DOCX)

**S7 Table. Results of multivariate regression analysis for DASS-21 scores of depression, anxiety, and stress for the participants in South America and Caribbean regions (n = 292).**
(DOCX)

**S8 Table. Results of multivariate regression analysis for DASS-21 scores of depression, anxiety, and stress for the participants in Oceania (n = 173).**
(DOCX)

## Acknowledgments

We thank all participants in this online survey study. We would like to also thank all members of Prof. Michelle L. Bell's research group for their contribution to the early drafts of the survey questionnaire in this research.

## Author Contributions

**Conceptualization:** Seulkee Heo, Michelle L. Bell.

**Data curation:** Seulkee Heo.

**Formal analysis:** Seulkee Heo.

**Funding acquisition:** Michelle L. Bell.

**Investigation:** Seulkee Heo.

**Methodology:** Seulkee Heo.

**Project administration:** Seulkee Heo, Michelle L. Bell.

**Resources:** Seulkee Heo, Pedro Diaz Peralta, Lan Jin, Claudia Ribeiro Pereira Nunes, Michelle L. Bell.

**Software:** Seulkee Heo.

**Supervision:** Seulkee Heo, Michelle L. Bell.

**Validation:** Seulkee Heo, Michelle L. Bell.

**Visualization:** Seulkee Heo.

**Writing – original draft:** Seulkee Heo, Michelle L. Bell.

**Writing – review & editing:** Seulkee Heo, Pedro Diaz Peralta, Claudia Ribeiro Pereira Nunes, Michelle L. Bell.

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
