## [Decision Letter · Decision Letter 0]

25 Apr 2022

PONE-D-22-10065Differences in self-perception of productivity and mental health among the STEMM-field scientists during the COVID-19 pandemic by sex and status as a parent: a survey in six languagesPLOS ONE

Dear Dr. Heo,

Thank you for submitting your manuscript to PLOS ONE. After careful consideration, we feel that it has merit but does not fully meet PLOS ONE’s publication criteria as it currently stands. Therefore, we invite you to submit a revised version of the manuscript that addresses the points raised during the review process. Please submit your revised manuscript by 22 May 2022. If you will need more time than this to complete your revisions, please reply to this message or contact the journal office at plosone@plos.org. Please include the following items when submitting your revised manuscript:A rebuttal letter that responds to each point raised by the academic editor and reviewer(s). You should upload this letter as a separate file labeled 'Response to Reviewers'.A marked-up copy of your manuscript that highlights changes made to the original version. You should upload this as a separate file labeled 'Revised Manuscript with Track Changes'.An unmarked version of your revised paper without tracked changes. You should upload this as a separate file labeled 'Manuscript'.If applicable, we recommend that you deposit your laboratory protocols in protocols.io to enhance the reproducibility of your results. Protocols.io assigns your protocol its own identifier (DOI) so that it can be cited independently in the future. For instructions see: https://journals.plos.org/plosone/s/submission-guidelines#loc-laboratory-protocols. Additionally, PLOS ONE offers an option for publishing peer-reviewed Lab Protocol articles, which describe protocols hosted on protocols.io. Read more information on sharing protocols at https://plos.org/protocols?utm_medium=editorial-email&utm_source=authorletters&utm_campaign=protocols.

We look forward to receiving your revised manuscript.

Kind regards,

Muhammad Shahzad Aslam, Ph.D.,M.Phil., Pharm-D

Academic Editor

PLOS ONE

Journal Requirements:

a) Did participants provide their written or verbal informed consent to participate in this study?

"The authors appreciate Yale Internal Women Faculty Forum (WFF) Seed Grant for supporting this research."

"This study was funded by Yale Internal Women Faculty Forum (WFF) Seed Grant awarded to Prof. Michelle L. Bell [available at https://wff.yale.edu/what-were-doing/research/seed-grants]. The funder had no role in study design, data collection and analysis, decision to publish, or preparation of the manuscript. "

Reviewers' comments:

Reviewer's Responses to Questions

**Comments to the Author**

1. Is the manuscript technically sound, and do the data support the conclusions?

Reviewer #1: Yes

Reviewer #2: Yes

2. Has the statistical analysis been performed appropriately and rigorously? 

Reviewer #1: Yes

Reviewer #2: Yes

3. Have the authors made all data underlying the findings in their manuscript fully available?

Reviewer #1: No

Reviewer #2: Yes

4. Is the manuscript presented in an intelligible fashion and written in standard English?

Reviewer #1: Yes

Reviewer #2: Yes

5. Review Comments to the Author

Reviewer #1: The manuscript is a technically sound piece of scientific research and very meticulously articulated. The methodological approaches are very comprehensive. The detailed step-by-step description of data collection and analyses is very well described. An important, comprehensive study with significant findings. The authors have declared that no competing interests exist. The online survey was totally anonymous no personal information was collected and moreover, there were no direct contacts between the participants and authors. The authors have stated that due to the nature of the research participants of the study did not agree for their data to be shared publicly therefore, supporting data will not be available.

Recommended for publishing with some very few spelling corrections.

Please correct the COVID-19 spelling on page 27 (line 428)

(line 457-458) should be “To the best of our knowledge” instead of “to our best of our knowledge

Reviewer #2: I want to congratulate the authors for this brilliant study.

I am happy to give it a go ahead with the following suggestions:

In general comment: I would advise the authorship to consider discussing the impacts of COVID-19 pandemic in general (in other sectors, across age groups, gender identities, different contexts) as the first paragraph of the introduction. After that, they may narrow down towards academic sector and then justify why it is important to highlight the impact of of COVID-19 on academic sector (In general). After that the next paragraph may introduce gender dimensions in academia and COVID-19 impact.

Specific Comment: Line 42 - : (As a reader) This sounds like until March 2020, they (WHO) were sitting on it, which is a debate and should not be triggered in our vocabulary. Suggestion: "The COVID-19 pandemic that caught the world's attention in (month, year) was declared a public health emergency of international concern by the WHO in Month, Date (Ref).

6. PLOS authors have the option to publish the peer review history of their article (what does this mean?). If published, this will include your full peer review and any attached files.

Reviewer #1: **Yes: **Raafat Hassan

Reviewer #2: **Yes: **Ateeb Ahmad Parray

---

## [Author Response · Author response to Decision Letter 0]

4 May 2022

Review Comments to the Author

Reviewer #1: The manuscript is a technically sound piece of scientific research and very meticulously articulated. The methodological approaches are very comprehensive. The detailed step-by-step description of data collection and analyses is very well described. An important, comprehensive study with significant findings. The authors have declared that no competing interests exist. The online survey was totally anonymous no personal information was collected and moreover, there were no direct contacts between the participants and authors. The authors have stated that due to the nature of the research participants of the study did not agree for their data to be shared publicly therefore, supporting data will not be available.

Recommended for publishing with some very few spelling corrections.

Please correct the COVID-19 spelling on page 27 (line 428)

(line 457-458) should be “To the best of our knowledge” instead of “to our best of our knowledge

A: Thank you for your comments. We corrected the spelling “COIVD-19” to “COVID-19” on page 21 line 371 and page 29 line 466 and line 471. The typo for “To our best of our knowledge” was changed to “To the best of our knowledge” on page 30 line 500.

Reviewer #2: I want to congratulate the authors for this brilliant study.

I am happy to give it a go ahead with the following suggestions:

In general comment: I would advise the authorship to consider discussing the impacts of COVID-19 pandemic in general (in other sectors, across age groups, gender identities, different contexts) as the first paragraph of the introduction. After that, they may narrow down towards academic sector and then justify why it is important to highlight the impact of COVID-19 on academic sector (In general). After that the next paragraph may introduce gender dimensions in academia and COVID-19 impact.

A: Thank you for your suggestion. We added new text about the impacts of COVID-19 on work environments in general and gender differences in work, poverty, and employment in the Introduction (please see page 3 lines 46-64). In page 4 lines 69-78, we added new text to introduce major changes in work routines in academia and the importance of studying challenges for scientific productivity and health in academia. We separated the original paragraph into two paragraphs and the current second paragraph (which starts on page 4 lines 79) introduces gender dimensions in academic fields. 

Specific Comment: Line 42 - : (As a reader) This sounds like until March 2020, they (WHO) were sitting on it, which is a debate and should not be triggered in our vocabulary. Suggestion: "The COVID-19 pandemic that caught the world's attention in (month, year) was declared a public health emergency of international concern by the WHO in Month, Date (Ref).

A: Thank you for your suggestion. Please see page 3 lines 42-44. We replaced the original sentence “The COVID-19 pandemic recognized in March 2020 by the World Health Organization” to the new text that reads “The COVID-19 pandemic that received international attention in (January 2020) was declared a public health emergency of international concern by the World Health Organization (WHO) in March 2020.”

---

## [Decision Letter · Decision Letter 1]

30 May 2022

Differences in self-perception of productivity and mental health among the STEMM-field scientists during the COVID-19 pandemic by sex and status as a parent: a survey in six languages

PONE-D-22-10065R1

Dear,

We’re pleased to inform you that your manuscript has been judged scientifically suitable for publication and will be formally accepted for publication once it meets all outstanding technical requirements.

Kind regards,

Muhammad Shahzad Aslam, Ph.D.,M.Phil., Pharm-D

Academic Editor

PLOS ONE

Additional Editor Comments (optional):

Reviewers' comments:

Reviewer's Responses to Questions

**Comments to the Author**

1. If the authors have adequately addressed your comments raised in a previous round of review and you feel that this manuscript is now acceptable for publication, you may indicate that here to bypass the “Comments to the Author” section, enter your conflict of interest statement in the “Confidential to Editor” section, and submit your "Accept" recommendation.

Reviewer #1: All comments have been addressed

Reviewer #2: All comments have been addressed

2. Is the manuscript technically sound, and do the data support the conclusions?

Reviewer #1: Yes

Reviewer #2: Yes

3. Has the statistical analysis been performed appropriately and rigorously? 

Reviewer #1: Yes

Reviewer #2: Yes

4. Have the authors made all data underlying the findings in their manuscript fully available?

Reviewer #1: No

Reviewer #2: Yes

5. Is the manuscript presented in an intelligible fashion and written in standard English?

Reviewer #1: Yes

Reviewer #2: Yes

6. Review Comments to the Author

Reviewer #1: (No Response)

Reviewer #2: Thank you for incorporating all the comments appropriately. The manuscript now describes a technically sound piece of scientific research with data that supports the conclusions. The conclusions have also been drawn appropriately based on the data presented.

7. PLOS authors have the option to publish the peer review history of their article (what does this mean?). If published, this will include your full peer review and any attached files.

Reviewer #1: **Yes: **Raafat Hassan

Reviewer #2: **Yes: **Ateeb Ahmad Parray

---

## [Editor Report · Acceptance letter]

23 Jun 2022

PONE-D-22-10065R1 

Differences in self-perception of productivity and mental health among the STEMM-field scientists during the COVID-19 pandemic by sex and status as a parent: a survey in six languages 

Dear Dr. Heo:

I'm pleased to inform you that your manuscript has been deemed suitable for publication in PLOS ONE. Congratulations! Your manuscript is now with our production department. 

Kind regards, 

on behalf of

Dr. Muhammad Shahzad Aslam 

Academic Editor

PLOS ONE